# OpenReview forum: "Efficient Recomputation of Marginal Likelihood upon Adding Training Data in Gaussian Processes and Simulator Fusion"
_ICLR.cc/2024/Conference — Submitted to ICLR 2024_

### Official Review · Reviewer_yLdo · 2023-10-30

**Soundness:** 2 fair
**Presentation:** 3 good
**Contribution:** 1 poor
**Rating:** 5
**Confidence:** 4

**Summary:**

This paper describes a method to incorporate extra training data to learn a particular prediction tasks using Gaussian processes. The proposed method consists in following a particular approach to decide whether a particular data instance should be incorporated to the training data or not. The criterion followed consists in using the negative log likelihood given by the predictive distribution of the GP after incorporating a particular training instance. This is equivalent to changing the prior GP to another GP that is expected to perform better (since it gives a better marginal likelihood estimation). The proposed method is expensive if a naive implementation is followed, with O(N^4) cost on the number of data points or iterations to follow. The authors proposed a clever implementation that takes into account partial updates of Cholesky factors. The method is validated on synthetic datasets both in terms of performance and in terms of computational cost.

**Strengths:**

The paper is very well written and clearly explained. Apart from that, I cannot find any other particular strength. My overall impression is that the paper is still in an early stage and needs more work before it can be accepted for publication. In particular, the authors should address the weaknesses described below.

**Weaknesses:**

The experimental section is too weak. It only considers synthetic datasets. It is not clear at all if the proposed method has a practical utility since no real world problems are considered in the paper. This questions the significance of the results. The authors should given particular examples of the expected utility of the proposed approach in a real-world setting.

        The paper lacks a solid related work section. It is not clear at all if this problem has been already studied in the literature and if some methods have already been devised for it. In the introduction there are some related methods described. However, they seem methods proposed for a different setting that may be adapted to the particular setting considered by the authors.

        The proposed method has a very large computational cost that is cubic w.r.t. to the number of training points or the points to be added to the training set. This is a limitation since only a few thousand points may be considered at most. The authors should try to scale the method to larger experimental settings, considering e.g., approaches for sparse GPs.

        The use of Cholesky factors that are updated efficiently is not new within the GP literature.

**Questions:**

Why do not each method in Fig. 1 start from the same initial value?

Could you approach be extended to take advantage of sparse GPs approaches to scale to large datasets?

---

> ### Author Response · Authors · 2023-11-21
> **Response to AnonReviewer yLdo**
>
> We're really grateful for the time and effort you've put into reviewing our work. Your constructive feedback has been very valuable to us.
>
> Firstly, we would like to address the concerns raised by all reviewers regarding the lack of experiments with real data. We agree that conducting experiments with real data is crucial for enhancing the reliability of our method. Therefore, we have conducted these experiments and included the results in Figure 1 (c). Please review the updated figure for your reference.
>
> Weaknesses
> > The experimental section is too weak. It only considers synthetic datasets.
>
> In response to the concern regarding the use of real datasets, we have conducted additional experiments with actual data. The methodology and the description of these experiments are detailed in the third paragraph of Section 4.1, and the results are presented in Figure 1(c). Although the differences are not as pronounced as with synthetic data, the results show an improvement in MSE over the standard Gaussian Process. Furthermore, they demonstrate that our approach is not influenced by deviations from the true distribution, which can affect methods like SoD when using simulator data.
>
> > The paper lacks a solid related work section.
>
> We appreciate the feedback regarding the related work section. Due to space constraints within the main text, we chose to briefly introduce only the most pertinent related works in the Introduction and provided a comprehensive review of the related literature in Appendix A. We acknowledge that the location of this section may not be immediately apparent to readers. To address this, we have now explicitly indicated in multiple relevant parts of the Introduction that a detailed discussion of related works can be found in Appendix A. This should guide the readers to the extended review, ensuring they are aware of the broader context and precedent for our work.
>
> > The proposed method has a very large computational cost that is cubic w.r.t. to the number of training points or the points to be added to the training set. This is a limitation since only a few thousand points may be considered at most. The authors should try to scale the method to larger experimental settings, considering e.g., approaches for sparse GPs.
>
> We acknowledge the reviewer's concerns regarding the computational expense of our method. As depicted in Figure 4, the computational time for our proposed method, even with 7000 training points, is under three hours. It's important to clarify that this computation is not recurrent for every prediction, but is a one-time cost incurred during the data selection phase from the simulator. Given this context, we consider the computational demand to be practical for the intended applications.
>
> > The use of Cholesky factors that are updated efficiently is not new within the GP literature.
>
> The efficient update of Cholesky factors is indeed a known technique within GP literature, as (\cite{osborne2010bayesian}) have previously demonstrated the computation of $\mathbf{L}\_{m+1}$ from $\mathbf{L}\_{m}$ in $O(m^2)$ time. Our method, however, introduces an efficient computation for deriving $\mathbf{V}\_{m+1}$ from $\mathbf{V}\_{m}$ with a complexity of $O(mN+N^2)$, where $\boldsymbol{\Sigma}\_{m+1}=\mathbf{V}\_{m+1}\mathbf{V}\_{m+1}^\mathsf{T}$. To communicate this effectively, we have detailed this contribution in footnote 5. We believe our computational technique, which is elaborated in Section 3.2, is not trivial.
>
> Questions:
>
> > Why do not each method in Fig. 1 start from the same initial value?
>
> The horizontal axis in Figure 1 represents the number of true training data points, not the generated data. The data point at the left end of Figure 1 corresponds to the MSE when there are 50 true training data points. Since each method selects a different number and subset of generated data to use, the MSEs vary accordingly. This explains the difference in starting points across the methods depicted in the figure.
>
> > Could you approach be extended to take advantage of sparse GPs approaches to scale to large datasets?
>
> Yes, our approach can indeed be extended to leverage sparse Gaussian processes for scalability to larger datasets. The most straightforward adaptation would be to first select the data from the simulator to be added to the training set using our proposed method, and then treat the combined dataset from the selected data and the original training data indistinctly when applying sparse GPs approaches. There is also potential for a more sophisticated integration of our method with sparse GPs, which could further enhance efficiency and scalability.

---

> > ### Comment · Reviewer_yLdo · 2023-11-22
> > **Response**
> >
> > I acknowledge the effort made by the reviewers to improve their paper, and I have increased a bit my score in consequence. However, including only one real-world problem in the paper, I think is insufficient. Therefore, I still think that this paper needs more work.

---

### Official Review · Reviewer_1Da7 · 2023-10-31

**Soundness:** 2 fair
**Presentation:** 2 fair
**Contribution:** 2 fair
**Rating:** 5
**Confidence:** 4

**Summary:**

This work proposes to use the negative log marginal likelihood of the Gaussian process as a criterion when selectively adding simulator-generated data to the training data. Since evaluating each candidate training data point using the negative log marginal likelihood can be time-consuming, the authors propose a method for fast computation by considering the so-called Cholesky update and take advantage of the dependencies between matrix elements.

**Strengths:**

Originality: Probably, the faster re-computation of the marginal likelihood might be the originality of the work.

Quality: The experiments provided in the paper are useful to provide an idea of the approach, though they are limited to just presenting synthetic scenarios.

Clarity: The methodology sections are generally well written and not difficult to follow, though they present some inconsistencies in the mathematical notation.

Significance (importance): The work has its strength in the efficient computation of the marginal likelihood.

**Weaknesses:**

-The idea of accepting data to be added as part of the training set by improving the marginal likelihood was previously explored by Titsias M. in "Variational Learning of Inducing Variables in Sparse Gaussian Processes" section 3.1. In the context of Titsias' work was used to generated pseudo-inputs (or inducing points).

-The experiments provided are limited to just presenting synthetic scenarios.

-The introduction does not properly motives the research problem to engage the reader with the work. Also, the introduction lacks of references to better support different phrases or claims. The methodology sections are generally well written and not difficult to follow, though they present some inconsistencies in the mathematical notation.

-The work has its strength in the efficient computation of the marginal likelihood, but the main aim of the work was not to compare such an algorithm with other approaches that improve such computation, but to introduce a direct method of selectively adding simulator-generated data to training data when using Gaussian processes.

**Questions:**

---Specific comments---

-In Abstract, it sounds contradictory to say that we rely on knowledge that is unreliable: "construct models based on their knowledge of the modeling target and, as training data increases, choose more flexible models with reduced dependence on that knowledge if that knowledge is unreliable."
Maybe the last sentence would be better understood if read as:"...if that knowledge becomes unreliable" or simply get rid of last part and leave: "construct models based on their knowledge of the modeling target and, as training data increases, choose more flexible models with reduced dependence on that knowledge."

-In Abstract, it is not clear what it is the intention of "We propose a faster method considering the Cholesky factor and matrix element dependencies." There is something missing to properly connect with all the previous text.

-In the Introduction, there is probably a sentence missing at the very beginning regarding modelling issues or modelling challenges than allows the reader understand where the idea or problem of bias-variance trade-off comes from. Also, it is necessary to include a strong reference regarding "bias-variance trade-off" to support the text.

-In the introduction, the phrase that reads: "On the other hand, the method of selectively adding generated
simulator data to the training data only requires that data can be generated from the simulator" seems ambiguous or needs rewording.

-Please include references to support: "The criterion for selecting important data is the diversity of the training data. Various methods to measure this
diversity have been proposed."

-Please include references to support: "The negative log marginal likelihood is a metric that measures the model’s ﬁt to the training data and has a theoretical foundation that it matches, on average, the KL divergence between the true distribution and the model’s distribution."

-Where it reads: "Within this category, although
Auto Data Augmentation is efficient, The knowledge transferred", lower case "..., The knowledge..." to "..., the knowledge..."

-Introduce the acronyms KL, BIC, GPs, NLL and NML!

In section 2.1:
-There seems to be inconsistency in the notation. I do not see the benefit of referring to $\mathbf{X}$ as a random variable. There is no information or specification of the distribution that $\mathbf{X}$ follows. I would suggest to refer as an input variable $\mathbf{x} \in \mathbb{R}^d$ instead of $\mathbf{X} \in \mathbb{R}^d$. Also $y \in \mathbb{R}$ instead of $\mathbf{y} \in \mathbb{R}^1$, these to be congruent with Eq. (1).

-Also, I suggest to use $\mathbf{y}^N=(y_1,y_2,...,y_N)^\top$ and $\mathbf{X}^N=(\mathbf{x}_1,\mathbf{x}_2,...,\mathbf{x}_N)^\top$ to be more consistent instead of the current notation in the paper.

In Eq. (1), the Covariance matrices $\mathbf{K}_{N,m^*}$

and $\mathbf{K}^{\top}_{N,m^*}$

might be swapped of quadrant.

It is more intuitive to think that the pair $N,m^*$ refers to rows,columns respectively.

Add period "." at the end of the equation.

Putting the $\mathbf{x}_1...\mathbf{x}_N$ and

$\mathbf{x}_{1^*}$ ...

$\mathbf{x}_{m^*}$

inside the equation looks strange as if a vector were multiplying the covariance matrix. Maybe a footnote should be added to avoid confusion.

-If $\mathbf{K}_{N,m^*}$

is swapped by $\mathbf{K}^{\top}_{N,m^*}$ then the equations that use these matrices should be corrected.

-Similar comment to the one before applies to Eq. (2).

-After Eq. (2) in $F_{m+1^*}$ the $y^{m+1}$ is missing "*".

-In Eq. (2) and (3) the identity matrices $\mathbf{I}$ should be different at each quadrant since they do not have the same dimensions.

In Eq (3) the negative sign is not applied, previously it was introduced $F_{m+1^*}=-\log \mathcal{N}...$. Also, as per Eq. (4) the operation in Eq. (3) should be

$(\mathbf{y}^{m+1}-\boldsymbol{\mu}_{m+1})$

instead of

$(\mathbf{y}^N-\boldsymbol{\mu}_{m+1})$

-Add a comma "," after Eq. (3) and (4), then period "." after Eq. (5).

-In section 3: write $(m+1)\times(m+1)$ instead of $m+1 \times m+1$. Indeed, in the equations should be better to write, say,

$\mathbf{y}_{(m+1)^*}$ or

$\mathbf{K}_{(m+1)^*}$.

-In section 3: it reads: "with a total cost of

$\mathcal{O}(M^2N + MN^2)$,

keeping it within the cubic order", shouldn't it be within the quadratic order?

-Before Eq. (6): what is $\mathbf{K}_{+m}$? typo?

-Before section 3.2:

$(\mathbf{L}_{m+1}$

$\mathbf{L}^{\top}_{m+1})^{-1}\mathbf{y}^{(m+1)^*}$

instead of

$(\mathbf{L}_{m+1}$

$\mathbf{L}^{\top}_{m+1})^{-1}\mathbf{y}^{m+1}$

-Where it reads: "Lalchand \& Faul
(2018) described in Section 1, promote diversity of training data." should be "promotes" since you are referring to the method or work.

-Typo where it reads: "then using the likelihood of the all output data y", should be "...of all the output data..."

-In section 4.2: "the number of training data candidates generated from the simulator was 1,000,", you mean "1000" or 1?

-In the figure 3: it is not possible to visualise the Training data (brown-ish colour) for SoD.

-In the conclusion: "the algorithm we proposed is specialized for regression models", not regression models in general, but a regression model particularly with a Gaussian likelihood.

-Generally, there is either a comma or period missing after the equations.

-Initial capital letter in the bibliography, words like: Gaussian and Cholesky.

-Why is there a distribution $q(\mathbf{X}^N)$ in appendix H for Eq. (21)? Aren't we saying in $KL(q(.|\mathbf{X}^N)||p(.|\mathbf{X}^N))$ that $\mathbf{X}^N$ is given? I do not think the Eq. (21) is correct.

---Other Questions---

Is this method feasible to different statistical data types for the outs $y$ or we should assume that $y$ is always in the real values?

We fit the GP hyperparameters with the training data, but are those hyperparameters tuned again when adding simulator data?

-The experiments shown seem to have an appropriate number of N data observation so that the GP model fits quite well for the range of input data $\mathbf{x}$, so due to the conditioning properties of a Normal distribution it is expected to only accept data that could improve the conditional distribution $p(\mathbf{y}^N|\mathbf{X}^N,\mathbf{y}^{m^*},\mathbf{X}^{m^*})$. What would it happen if the GP has a smaller number of data observation, or lack of data in regions such that the predictive distribution was less uncertain? How would the acceptance and rejection would behave in such a region?

-It seems that the Log marginal likelihood metric gives priority to the model fitting, so when do we trust the simulator?

-What if the simulator is actually quite close to the true distribution, but we have a small number of data observations for which we fit a GP with the hyper-parameters tuning a distribution not that close to the true distribution?


What ways to measure a trade-off, as mentioned in the introduction, to achieve an appropriate bias-variance in our last model that contains training and simulator data?

-If I fit the GP and generate data from such a GP and use it as simulator data, wouldn't I expect to achieve improvements in the Log marginal likelihood? Wouldn't the fitted GP be simply the best data simulator?

-The work is missing to show a real world application to additionally assess the performance of the approach, for instance an example as claimed in appendix C.

-What would be the effect of using different data simulators? For instance, a simulator less similar to the real distribution.

-For the practitioner, How is a data simulator generally built or where does it come from?

-What if the dataset we are fitting presents a heteroscedastic noise, how could this affect the method approach for accepting training data candidates?

---

> ### Author Response · Authors · 2023-11-21
> **Response to AnonReviewer 1Da7 (1)**
>
> We appreciate the detailed feedback you provided. Your expertise has played a key role in making our paper clearer and better.
>
> Firstly, we would like to address the concerns raised by all reviewers regarding the lack of experiments with real data. We agree that conducting experiments with real data is crucial for enhancing the reliability of our method. Therefore, we have conducted these experiments and included the results in Figure 1 (c). Please review the updated figure for your reference.
>
> Weaknesses:
> > The idea of accepting data to be added as part of the training set by improving the marginal likelihood was previously explored by Titsias M. in "Variational Learning of Inducing Variables in Sparse Gaussian Processes" section 3.1. In the context of Titsias' work was used to generated pseudo-inputs (or inducing points).
>
> I had not noticed that in the paper. Thank you for pointing it out.
> While it is true that (Titsias2009a) utilizes the lower bound of the marginal likelihood as a metric, our method differs in that it directly employs the marginal likelihood itself. (Titsias2009a) aimed to develop sparse GPs for reducing training data by replacing the dependence of function values $\mathbf{f}$ on the training data $\mathbf{X}$ with $p(\mathbf{f}|\mathbf{f}^m)$, thus altering the model's marginal likelihood from Equation 4 in (Titsias2009b) to Equation 10 in the same work. Consequently, the marginal likelihood could not be quickly computed, leading to the use of its lower bound as a metric (as shown in Equation 8 in (Titsias2009a) or Equation 13 in (Titsias2009b)). In contrast, our proposed application allows for the use of all training data, thereby eliminating the need to modify the marginal likelihood. Our acceleration method enables the use of the log likelihood itself as the metric. We have added (Titsias2009a) and other papers employing log marginal likelihood to the related works in Appendix A.2 and clarified these differences as described above.
>
> > The experiments provided are limited to just presenting synthetic scenarios.
>
> In response to the concern regarding the use of real datasets, we have conducted additional experiments with actual data. The methodology and the description of these experiments are detailed in the third paragraph of Section 4.1, and the results are presented in Figure 1(c). Although the differences are not as pronounced as with synthetic data, the results show an improvement in MSE over the standard Gaussian Process. Furthermore, they demonstrate that our approach is not influenced by deviations from the true distribution, which can affect methods like SoD when using simulator data.
>
> > The introduction does not properly motives the research problem to engage the reader with the work. Also, the introduction lacks of references to better support different phrases or claims. The methodology sections are generally well written and not difficult to follow, though they present some inconsistencies in the mathematical notation.
>
> Thank you for your detailed feedback regarding the Abstract, Introduction, and Methodology sections as outlined in the Questions section. We have taken your comments into careful consideration and made appropriate revisions to address these concerns.
>
> > The work has its strength in the efficient computation of the marginal likelihood, but the main aim of the work was not to compare such an algorithm with other approaches that improve such computation, but to introduce a direct method of selectively adding simulator-generated data to training data when using Gaussian processes.
>
> Our study represents the first instance of employing log marginal likelihood for data selection in Gaussian Processes, distinguishing it from (Titsias2009a) which uses the lower bound of log marginal likelihood. This novelty means that there have been no previous algorithms developed for efficiently computing this specific application. Given the unique nature of our approach, creating alternative algorithms that similarly reduce computational complexity is a non-trivial task. Hence, the main contribution of our work is not just in the efficient computation of marginal likelihood, but also in its innovative use for incorporating simulator-generated data into training sets in Gaussian Processes.
>
> (Titsias2009a) Titsias, Michalis. "Variational learning of inducing variables in sparse Gaussian processes." Artificial intelligence and statistics. PMLR, (2009).
>
> (Titsias2009b) Titsias, Michalis K. "Variational model selection for sparse Gaussian process regression." Report, University of Manchester, UK (2009).

---

> ### Author Response · Authors · 2023-11-21
> **Response to AnonReviewer 1Da7 (2)**
>
> Questions:
>
> Thank you for the detailed review of the equations and for pointing out typos. We have essentially corrected all the equations and typos. Specifically, the corrections that were highlighted in your comments are presented below.
>
> > In Abstract, it sounds contradictory to say that we rely on knowledge that is unreliable: "construct models based on their knowledge of the modeling target and, as training data increases, choose more flexible models with reduced dependence on that knowledge if that knowledge is unreliable." Maybe the last sentence would be better understood if read as:"...if that knowledge becomes unreliable" or simply get rid of last part and leave: "construct models based on their knowledge of the modeling target and, as training data increases, choose more flexible models with reduced dependence on that knowledge."
>
> Thank you for your insightful feedback and specific suggestions regarding the phrasing in the Abstract. You are correct in pointing out that the term "unreliable" may not be the most appropriate, as the context is not limited to situations where the knowledge is incorrect. We have taken your advice and revised the sentence to the second option you proposed: "construct models based on their knowledge of the modeling target and, as training data increases, choose more flexible models with reduced dependence on that knowledge.
>
> > In Abstract, it is not clear what it is the intention of "We propose a faster method considering the Cholesky factor and matrix element dependencies."
>
> We appreciate the feedback on the clarity of our abstract. To clarify the motivation for developing a faster method, we have revised the text as follows: "On the other hand, the log marginal likelihood is a theoretically grounded metric when viewed as a model selection criterion for incorporating data generated from a simulator into the training data. Calculating this metric for GPs is computationally expensive due to the necessity of inverting large covariance matrices. To address this challenge, our proposed method accelerates the process by efficiently updating the Cholesky decomposition and considering the dependencies between matrix elements. This improvement not only expedites computations but also maintains the theoretical robustness of the model selection process."
>
> > In the Introduction, there is probably a sentence missing at the very beginning regarding modelling issues or modelling challenges than allows the reader understand where the idea or problem of bias-variance trade-off comes from. Also, it is necessary to include a strong reference regarding "bias-variance trade-off" to support the text.
>
> Thank you for the constructive feedback. Indeed, the initial motivation for considering the bias-variance trade-off was not clearly stated. To rectify this, we have added an introductory sentence to the Introduction that outlines the general aim of regression models, smoothly transitioning to the discussion of the bias-variance trade-off. The revised text is as follows:
>
> "One of the ultimate objectives of machine learning models is to reduce generalization loss. According to the bias-variance trade-off, the generalization loss, when employing Mean Squared Error (MSE) as the loss function, can be decomposed into terms of bias and variance."
>
> In addition, we have simplified the explanation of the bias-variance trade-off and have incorporated a strong reference to support the concept by adding PRML (Bishop2006), which discusses the same in the context of Equation 3.44.
>
> (Bishop2006) Christopher M Bishop and Nasser M Nasrabadi. Pattern recognition and machine learning. Springer, 2006.
>
> > In the introduction, the phrase that reads: "On the other hand, the method of selectively adding generated simulator data to the training data only requires that data can be generated from the simulator" seems ambiguous or needs rewording.
>
> Indeed, the sentence in question could have been misinterpreted due to its redundant structure. We have revised it for clarity and conciseness as follows:
> "On the other hand, the method of selectively adding generated data requires only the ability to produce data from the simulator."
>
> > Please include references to support: "The criterion for selecting important data is the diversity of the training data. Various methods to measure this diversity have been proposed."
>
> We have included relevant references of "the various methods".
>
> > Please include references to support: "The negative log marginal likelihood is a metric that measures the model’s ﬁt to the training data and has a theoretical foundation that it matches, on average, the KL divergence between the true distribution and the model’s distribution."
>
> We have included references that detail the relationship between the negative log marginal likelihood and the KL divergence. Additionally, we offer a thorough explanation in Appendix G, H.

---

> ### Author Response · Authors · 2023-11-21
> **Response to AnonReviewer 1Da7 (3)**
>
> > looks strange as if a vector were multiplying the covariance matrix. Maybe a footnote should be added to avoid confusion.
>
> The following content has been added to Footnote 3 as indicated. Thank you for bringing this to our attention. $\mathbf{x}\_{1^*}\dots\mathbf{x}\_{m^*}, \mathbf{x}\_{1}\dots\mathbf{x}\_{N}$ represent the input variables for the kernel function that constructs the covariance matrix.
>
> > In Eq. (2) and (3) the identity matrices should be different at each quadrant since they do not have the same dimensions.
>
> The content has been added to Footnote 4 as indicated. Thank you for bringing this to our attention.
> "Different-sized identity matrices appear in the paper, but the size is easily inferred from the context, so they are all uniformly denoted as $\mathbf{I}$."
>
> > Also, as per Eq. (4) the operation in Eq. (3) should be $(\mathbf{y}^{m+1}-\mathbf{\mu}\_{m+1})$ instead of $(\mathbf{y}^{N}-\mathbf{\mu}\_{m+1})$.
>
> Indeed, the notation $\mathbf{y}^N$ as presented is accurate. To clarify the distribution aspect of $F\_{m+1^*}$ with respect to $\mathbf{y}^N$, we have now explicitly stated in the preceding sentence that $F_{m+1^*} = -\log p(\mathbf{y}^N|\ldots)$. This amendment should eliminate any confusion regarding the distribution of $\mathbf{y}^N$ in relation to $F_{m+1^*}$.
>
> > Indeed, in the equations should be better to write, say $\mathbf{K}\_{(m+1)^*}$.
>
> Acknowledging your point, we have decided not to add parentheses around $\mathbf{K}_{N, (m+1)^*}$ to avoid cluttering the notation. The current format should be clear enough to avoid any misunderstanding.
>
> > In the figure 3: it is not possible to visualise the Training data (brown-ish colour) for SoD.
>
> Indeed, due to the order in which the data points were plotted, the Training data (in a brown-ish color) for SoD was obscured in Figure 3. We have corrected this issue and adjusted the plotting order to ensure that all points are now clearly visible in the figure. Thank you for pointing out this oversight.
>
> > In the conclusion: "the algorithm we proposed is specialized for regression models", not regression models in general, but a regression model particularly with a Gaussian likelihood.
>
> Indeed, our algorithm is tailored for regression models where the observation noise is Gaussian. To make this clear, we have explicitly stated this limitation in the conclusion as follows:
> "As a limitation, the algorithm we proposed is specialized for regression models with Gaussian likelihood, and its extension to classification models and other likelihood models is not straightforward."
>
> > Why is there a distribution $q(\mathbf{X}^N)$ in appendix H for Eq. (21)? Aren't we saying in $KL(q(\cdot| \mathbf{X}^N || p(\cdot | \mathbf{X}^N)$ that $\mathbf{X}^N$ is given? I do not think the Eq. (21) is correct.
>
> We require $q(\mathbf{X}^N)$ to ensure that the distance between our model $p(\mathbf{y}^N|\mathbf{X}^N, \mathbf{X}^{m*}, \mathbf{y}^{m*})$ and the true distribution $q(\mathbf{y}^N|\mathbf{X}^N)$ is minimized on average over the true distribution $q(\mathbf{X}^N)$, rather than for a specific training dataset $\mathbf{X}^N$. This definition aligns with the conditional KL divergence as defined in (Poczos2012) Definition 5, which includes $q(\mathbf{X}^N)$. To avoid confusion, we have made the following modifications:
>
> - Cited (Poczos2012) right after mentioning "Conditional KL divergence".
> - Revised the description before Equation 20 from $q(\mathbf{y}^N|\mathbf{X}^N)$ to $q(\mathbf{y}^N|\mathbf{X}^N)q(\mathbf{X}^N)$ to reflect the true distribution of the dataset.
> - Adjusted Equations 20 and 21 to align with (Poczos2012)’s definition by bringing the integral over $\mathbf{X}^N$ to the outermost layer.
>
> (Poczos2012) Barnabas Poczos and Jeff Schneider. Nonparametric estimation of conditional information and divergences. In Artificial Intelligence and Statistics, 2012.
>
> > Is this method feasible to different statistical data types for the outs or we should assume that is always in the real values?
>
> The specific computational techniques we propose are tailored for scenarios where the outputs are in real values. In cases like classification problems, where the outputs take a different form, the formula for marginal log likelihood (as in Equation 3) changes, making our proposed computational techniques inapplicable.

---

> ### Author Response · Authors · 2023-11-21
> **Response to AnonReviewer 1Da7 (4)**
>
> > We fit the GP hyperparameters with the training data, but are those hyperparameters tuned again when adding simulator data?
>
> The hyperparameters of the Gaussian Process are determined from the initial training data and are not modified post-learning.
> "The hyperparameters of the input kernel function are pre-optimized using the initial training data $(\mathbf{X}^N, \mathbf{y}^N)$ and remain fixed throughout the execution of Algorithm 1. While re-optimization of the kernel's hyperparameters using the selected generated data $(\mathbf{X}^{M^*}, \mathbf{y}^{M^*})$ in conjunction with the original training data $(\mathbf{X}^N, \mathbf{y}^N)$ could be considered after the selection process, this was not performed in the current study."
> Further, while an iterative approach to hyperparameter optimization in conjunction with data selection, as suggested by (titsias2009), could be contemplated, such an exploration is reserved for future work.
>
> > The experiments shown seem to have an appropriate number of N data observation so that the GP model fits quite well for the range of input data , so due to the conditioning properties of a Normal distribution it is expected to only accept data that could improve the conditional distribution. What would it happen if the GP has a smaller number of data observation, or lack of data in regions such that the predictive distribution was less uncertain? How would the acceptance and rejection would behave in such a region?
>
> In regions where the true training data is sparse, adding generated data to the training set is likely to have a minimal impact on the predictions for true training data points that are distant from the added data, due to the properties of kernel methods. Consequently, the value of $F_{m+1}$is expected to change very little, whether it increases or decreases, compared to $F_m$. Therefore, the inclusion of such generated data in these sparse areas would likely be random. We have not conducted experiments specifically to support this hypothesis.
>
> > It seems that the Log marginal likelihood metric gives priority to the model fitting, so when do we trust the simulator?
>
> When it comes to prioritizing data over domain knowledge, our method indeed favors the log marginal likelihood, or the data. As mentioned in the Introduction, model assumptions can be adjusted automatically from data, which inherently places priority on the data. Trust in the simulator would require manual intervention to incorporate domain knowledge into the model assumptions. For instance, one might manually add all data from a trusted simulator region to the training set.
>
> > What if the simulator is actually quite close to the true distribution, but we have a small number of data observations for which we fit a GP with the hyper-parameters tuning a distribution not that close to the true distribution?
>
> This is indeed a fundamental question. Our approach could be seen as expanding the hyperparameter space of GPs. The selection of simulator-generated data to include in training acts as a hyperparameter, which introduces the risk of overfitting. If the simulator closely approximates the true distribution, this could be considered as constraining the hyperparameter space to a range closer to the correct values. For instance, the region represented by the gray and pink plots in Figure 3 could be seen as this hyperparameter space. Fitting hyperparameters always carries the risk of overfitting, particularly with a small number of true training data points. However, if the simulator matches the true distribution accurately, any selected generated data is likely to improve predictive performance.
>
> > What ways to measure a trade-off, as mentioned in the introduction, to achieve an appropriate bias-variance in our last model that contains training and simulator data?
>
> Bias-variance trade-off cannot be directly measured because we do not have knowledge of the true underlying distribution. Instead, we rely on indirect metrics. One such metric is the log marginal likelihood, which assesses the model's fit to the data and the balance of complexity (as discussed in Bishop, 2006, Section 3.4). Optimizing the log marginal likelihood indirectly contributes to achieving a favorable balance between bias and variance. We have added the motivation for using log marginal likelihood to the 6th paragraph of the Introduction to provide a clearer explanation.

---

> ### Author Response · Authors · 2023-11-21
> **Response to AnonReviewer 1Da7 (5)**
>
> > If I fit the GP and generate data from such a GP and use it as simulator data, wouldn't I expect to achieve improvements in the Log marginal likelihood? Wouldn't the fitted GP be simply the best data simulator?
>
> This is an intriguing consideration. The improvement of log marginal likelihood using data generated from a GP fitted to a limited dataset is not a given. For outlier values of $\mathbf{x}$ that are less likely to be sampled from the true distribution $q(y∣\mathbf{x})q(\mathbf{x})$, whether the generated data pairs $(\mathbf{x},y)$ from the GP's predictive distribution will enhance the log marginal likelihood is uncertain.
>
> > The work is missing to show a real world application to additionally assess the performance of the approach, for instance an example as claimed in appendix C.
>
> In response to the concern regarding the use of real datasets, we have conducted additional experiments with actual data. The methodology and the description of these experiments are detailed in the third paragraph of Section 4.1, and the results are presented in Figure 1(c). Although the differences are not as pronounced as with synthetic data, the results show an improvement in MSE over the standard Gaussian Process. Furthermore, they demonstrate that our approach is not influenced by deviations from the true distribution, which can affect methods like SoD when using simulator data.
>
> > What would be the effect of using different data simulators? For instance, a simulator less similar to the real distribution.
>
> We have investigated the effect of using simulators that diverge from the true distribution in Section 4.2, as depicted in Figure 2. The parameter $a$ increases the dissimilarity between the simulator's distribution and the true distribution. Our findings indicate that even as the simulator becomes less similar to the real distribution, the MSE of our proposed method remains relatively stable.
>
> > For the practitioner, How is a data simulator generally built or where does it come from?
>
> Practitioners generally build data simulators by translating the knowledge they currently possess about the relationship between y and x into a model. Our method is applicable as long as the simulator can generate pairs of $(y,x)$, regardless of whether the simulator describes the $(y,x)$ relationship analytically or generates $(y,x)$ hrough numerical computation.
>
> > What if the dataset we are fitting presents a heteroscedastic noise, how could this affect the method approach for accepting training data candidates?
>
> Process model assumes homoscedastic noise, so employing it on data with heteroscedastic noise implies using an incorrect model. In areas where the heteroscedastic noise variance is less than that assumed by our GP model, simulator data with variability within this range may not be sufficiently penalized in terms of the negative log marginal likelihood, potentially leading to incorrect acceptance. Conversely, in areas where the heteroscedastic noise variance is greater, our method might overly penalize simulator-generated data, resulting in excessive rejection.

---

> > ### Comment · Reviewer_1Da7 · 2023-11-22
> >
> > Thanks to the authors for the constructive discussion and for the different responses to my questions.

---

### Official Review · Reviewer_q1Av · 2023-11-09

**Soundness:** 3 good
**Presentation:** 3 good
**Contribution:** 2 fair
**Rating:** 6
**Confidence:** 3

**Summary:**

This paper focuses on selectively adding data for training a low-variance model, which is an important topic. The authors propose to deploy GP along with marginal likelihood as the metric to evaluate the quality of simulated data samples. The paper first talks about the method to selectively add more training data using GP. Then, it introduces the algorithm for faster implementation.  The experiments show the improvement.

**Strengths:**

- The GP for adding simulated data seems to better perform than other baseline methods.
- The algorithm is faster.
- The discussion is well-rounded.

**Weaknesses:**

- The novelty of GP on this topic is a bit limited. GP is not a new method at all. The algorithm that makes it faster is more interesting but no major breakthrough.
- It seems no real data set is experimented.

**Questions:**

1. Why gray points are not adopted in Figure 3?
2. It is said that the hyperparameters of GP are learned from initial training data. Do those hyperparameters change after it is learned? If it is not, does the initial training data affect the selection process? If it is not, how does it change?

---

> ### Author Response · Authors · 2023-11-21
> **Response to AnonReviewer q1Av**
>
> We want to thank the reviewer for the insightful comments and suggestions. Your advice has been a great help in improving our manuscript.
>
> Firstly, we would like to address the concerns raised by all reviewers regarding the lack of experiments with real data. We agree that conducting experiments with real data is crucial for enhancing the reliability of our method. Therefore, we have conducted these experiments and included the results in Figure 1 (c). Please review the updated figure for your reference.
>
> Weaknesses:
> > The novelty of GP on this topic is a bit limited. GP is not a new method at all. The algorithm that makes it faster is more interesting but no major breakthrough.
>
> As discussed in the third paragraph of the Introduction, Gaussian Processes (GPs) offer a unique advantage in incorporating the evolution of deep learning by allowing the kernel function to be represented as a neural network architecture. This adaptability ensures that GPs remains a relevant and powerful tool in the field. Moreover, the ability of GP to quantify predictive uncertainty is a significant contribution that enhances the capabilities of deep learning models. Hence, we contend that GP continues to be of substantial importance and relevance in our domain.
>
> > It seems no real data set is experimented.
>
> In response to the concern regarding the use of real datasets, we have conducted additional experiments with actual data. The methodology and the description of these experiments are detailed in the third paragraph of Section 4.1, and the results are presented in Figure 1(c). Although the differences are not as pronounced as with synthetic data, the results show an improvement in MSE over the standard Gaussian Process. Furthermore, they demonstrate that our approach is not influenced by deviations from the true distribution, which can affect methods like SoD when using simulator data.
>
>
> Questions:
> > Why gray points are not adopted in Figure 3?
>
> Thank you for bringing this to our attention. The omission of the gray points from the legend in Figure 3 was an oversight. We have now updated the figure to include these points in the legend.
>
> > It is said that the hyperparameters of GP are learned from initial training data. Do those hyperparameters change after it is learned? If it is not, does the initial training data affect the selection process? If it is not, how does it change?
>
> The hyperparameters of the Gaussian Process are determined from the initial training data and are not modified post-learning. These pre-learned hyperparameters, along with the free energy, influence the selection process through their interaction with the initial training data. Specifically, the computation of free energy is dependent on the initial training data's outputs, $\mathbf{y}^N$ (as defined in equation 2), and inputs, $\mathbf{x}_N$ (through $\mathbf{K}_N$ and $\mathbf{K}\_{N,m+1^*}$ as in equation 5). We have included additional clarification in Appendix B as follows:
>
> "The hyperparameters of the input kernel function are pre-optimized using the initial training data $(\mathbf{X}^N, \mathbf{y}^N)$ and remain fixed throughout the execution of Algorithm 1. While re-optimization of the kernel's hyperparameters using the selected generated data $(\mathbf{X}^{M^*}, \mathbf{y}^{M^*})$ in conjunction with the original training data $(\mathbf{X}^N, \mathbf{y}^N)$ could be considered after the selection process, this was not performed in the current study."
>
> Further, while an iterative approach to hyperparameter optimization in conjunction with data selection, as suggested by (Titsias2009), could be contemplated, such an exploration is reserved for future work.
>
> (Titsias2009) Titsias, Michalis. "Variational learning of inducing variables in sparse Gaussian processes." Artificial intelligence and statistics. PMLR, 2009.

---

### Meta-Review · Area_Chair_aRMK · 2023-12-11

**Metareview:**

The paper presents a way of using the marginal likelihood of a GP to generate simulated data in order to supplement a data set. The paper received three reviews, with two tending towards reject and one towards accept, but all reviewers felt the contribution was fair or poor. In particular, it was not clear how this work fits into previous approaches and the sense was that the novelty of the method was low and the experiments emphasized proof-of-concept on simulated data too much rather than its real world usefulness. Overall, this was thought to be a work in process that requires more development before it can be accepted to a conference like ICLR.

**Justification For Why Not Higher Score:**

The scores tended towards reject and no reviewer seemed convinced the paper represents a significant development of existing techniques.

**Justification For Why Not Lower Score:**

NA

---

### Decision · Program_Chairs · 2024-01-16

Reject